

# Biochemical changes throughout early- and middle-stages of embryogenesis in lobsters (*Homarus americanus*) under different thermal regimes

Jason S. Goldstein[1,2] and Winsor H. Watson III[2]

[1] Maine Coastal Ecology Center, Wells National Estuarine Research Reserve, Wells, ME, USA
[2] Department of Biological Sciences and School of Marine Sciences and Ocean Engineering, University of New Hampshire, Durham, NH, USA

## ABSTRACT

Most marine crustacean eggs contain the full complement of nutritional resources required to fuel their growth and development. Given the propensity of many ovigerous (egg-bearing) American lobsters (*Homarus americanus*) to undergo seasonal inshore-to-offshore migrations, thereby potentially exposing their eggs to varying thermal regimes, the goal of this study was to determine the impact of water temperature on egg quality over their course of development. This was accomplished by documenting changes in total lipids, proteins, and size (volume) of eggs subjected to one of three thermal regimes: inshore, offshore, and constant (16 °C) conditions. Total egg lipids showed a marked decrease over time, while protein levels increased over the same period. Although there were no significant differences in total lipids, proteins, or egg sizes between eggs exposed to inshore and offshore temperatures, they differed from values for eggs exposed to a constant temperature, which also hatched almost three months sooner. This is most likely due to the fact that eggs held at a constant temperature did not experience a period of slow development during the colder months from November to March that are important for synchronizing egg hatch and may be compromised by elevated seawater temperatures.

## INTRODUCTION

Egg development for most marine crustaceans relies heavily on the production and sequestering of nutrients required for the entire process of embryogenesis. In terms of biochemical constituents, both lipids and proteins play pivotal roles throughout development, and as a result, have been studied extensively in crustaceans and fishes (*Fraser, 1989*; *Jaeckle, 1995*; *Rosa et al., 2007*). Lipids comprise the structural integrity of most cells and are required for much of the metabolism of crustacean embryos, accounting for upwards of 60% of the total energy expenditure for growth (*Holland, 1978*; *Amsler & George, 1984*). By contrast, proteins constitute the basic building blocks of animal

Corresponding author
Jason S. Goldstein,
jgoldstein@wellsnerr.org,
jsgoldstein2@gmail.com

tissues (*Holland, 1978*), and may also function as alternative energy sources under certain conditions (*Schmidt-Nielsen, 1991*; *Heras, Gonzales-Baro & Pollero, 2000*).

Egg development in crustaceans is also very sensitive to thermal conditions, and incubation periods can be extended by cold water temperatures, or reduced during warmer conditions, thus influencing timing to hatch (reviewed in *Giménez, 2006*). Furthermore, as metabolic rates increase at higher temperatures, nutrients are used up at a faster rate and this can influence egg or larval condition (*Pandian, 1970*; *Schmidt-Nielsen, 1991*). Because most marine species' larvae derive their nutrition from both exogenous (feeding) and endogenous (yolk reserves) sources (*Sasaki, McDowell Capuzzo & Biesiot, 1986*) this can also have an impact on their development and metamorphosis. Therefore, the relationship between the primary biochemical components in crustacean eggs and their associated thermal variability are considered central to the early-life history patterns for these organisms (*Vance, 1973*; *Jaeckle, 1995*).

American lobsters, *Homarus americanus* H. Milne-Edwards (1837) are large, highly mobile decapods whose habitats include coastal and continental shelf waters, bays, and estuaries from Labrador, Canada to Cape Hatteras, US (*Fogarty, 1995*). Lobster eggs are typically extruded and fertilized in the late summer and fall, and then carried on the underside of the female's abdomen for 9–11 months before they hatch the following spring/summer. During this time, they are exposed to seasonal fluctuations in water temperature that can have a pronounced impact on their development (*Perkins, 1972*; *Goldstein & Watson III, 2015a*). Moreover, many ovigerous females undertake inshore-to-offshore seasonal migrations that can either enhance, or reduce, the magnitude of the seasonal temperature fluctuations their eggs experience (*Campbell & Stasko, 1986*; *Cowan et al., 2007*; *Goldstein & Watson III, 2015a*). For example, ovigerous lobsters subjected to inshore thermal regimes in laboratory-based studies exhibited more rapid egg development and hatched sooner than their offshore counterparts (*Goldstein & Watson III, 2015b*). This was most likely due to the rapid warming of the inshore waters in the spring, because the mean temperatures in the two locations were not significantly different. Therefore, *Goldstein & Watson III (2015a)* concluded that the seasonal offshore movements of ovigerous lobsters was most likely a strategy to enhance the survival of larvae, rather than a mechanism to speed up or delay egg development.

Although optimal temperatures for lobster egg growth are not fully known, naturally fluctuating temperatures result in disparate growth patterns and subsequently, differing hatch times (*Sibert, Ouellet & Brethes, 2004*; *Goldstein & Watson III, 2015b*). In general, either prolonged warm or cold temperatures can have a deleterious effect on the utilization of egg yolk reserves (*Garcia-Guerrero, Racotta & Villareal, 2003*; *Manush et al., 2006*), and it has been suggested that prolonged cold temperatures (<4 °C) negatively affect egg development in *H. americanus* (*Waddy & Aiken, 1995*).

The goal of this study was to determine the effects of temperature on the protein and lipid reserves of *Homarus americanus* eggs. Given the tendency for lobsters along the southern Maine coastline to migrate offshore in the winter, we hypothesized that this behavior might expose them to a thermal regime that maximized their metabolic efficiency, allowing for both optimal growth and maintenance of energy reserves in the form of stored lipids.

To test this hypothesis, we held females carrying fertilized eggs under different thermal regimes and monitored changes in total lipids, proteins, and egg size during the course of their development.

## METHODS

### Lobster source and egg assessment

A total of 15 egg-bearing (ovigerous) lobsters were legally collected (New Hampshire Fish & Game permit, RSA 214:29) in late August and early September (2006) along the New Hampshire (NH) seacoast near Rye, NH and Gunboat Shoals (43°.0274N; 70°.6938W, southern Gulf of Maine) by permitted commercial lobstermen using standard baited traps. Lobsters were transported to the University of New Hampshire (UNH) Coastal Marine Laboratory in Newcastle, NH and lobsters were initially held in a large 1,200 L fiberglass tank with shelters. The holding tank was sourced by ambient sand-filtered seawater (average temp = 15.3 ± 0.5 °C) and was subjected to ambient light. Lobsters in the tank were fed a combination of fresh squid and crabs (*Cancer spp.*), twice per week.

Lobster carapace lengths (CL) were measured to the nearest 1 mm using digital calipers (Mitutoyo IP 65, Mitutoyo Corp., Japan) (size range: 84–96 mm CL) and a laminated disc tag (diameter = 2.0 cm, Floy Tag Inc., Seattle, WA) was fastened to the propus of each animal with a single ziptie for identification throughout the duration of the study. A subset of eggs from each lobster clutch ($n$ =15–20) were viewed under a dissecting scope on September-5 and staged according to the methods outlined by *Helluy & Beltz (1991)*. Only lobsters whose clutches had eye indices <18% (*Perkins, 1972*; *Goldstein & Watson III, 2015b*) were used for this sample to encompass as much of the early development process as possible (*Perkins, 1972*).

### Thermal treatments and sampling

The experimental setup and thermal treatments are described in detail in *Goldstein & Watson III (2015b)*. Briefly, six 0.91 m diameter (600 L) tanks (2 tanks/ treatment) were used to mimic either inshore, offshore, or constant (16 ± 0.4 °C) temperature regimes on a year-round basis. Lobsters in each tank were kept isolated using mesh dividers and each animal received a shelter and was exposed to ambient photoperiod. For the purposes of this study thermal regimes were simulated to either match inshore locations (shallow and coastal, 2–5 km from shore, 8–10 m depth) or the offshore habitats (12–20 km from shore, 20–30 m depth) to which lobsters move to in the fall and overwinter (see *Goldstein & Watson III, 2015b*). Constant temperatures were chosen to simulate a favorable (and predictable) growth temperature similar to eggs observed in *MacKenzie (1988)*. Temperatures in all tanks were logged every 30 minutes using HOBO pendant loggers (model UA-002-64, Onset Computer, Bourne, MA) and later downloaded using Hoboware software (HOBOware Pro v. 3.0). Inshore temperatures mirrored the ambient seawater that was circulated through the CML seawater system while the temperatures in the offshore tanks were adjusted biweekly to simulate offshore temperatures based on historical and real time data published on the Northeastern Regional Association of Coastal Ocean Observing Systems website (NERACOOS, http://neracoos.org). A total of

five ovigerous females were exposed to each of the three temperature treatments and their eggs were sampled at five discrete time periods: twice in the fall (Oct-15 & Nov-15) during initial growth, once in the winter (Jan-15), when eggs remain in a relative stasis, and twice in the spring (Mar-15 & May-25), during periods of rapid growth (*Sibert, Ouellet & Brethes, 2004*; *Goldstein & Watson III, 2015b*).

Although in this study each lobster was not truly independently segregated, we chose this design for two reasons: (1) as experimental replicates, lobster eggs are endowed with very thick-layered egg casings that make them isolated from other eggs (with the exception of hatching events when the chorion essentially ruptures, (*Talbot & Helluy, 1995*); and (2) logistically, we were not able to isolate each lobster on 15 separate incoming seawater lines. This would have been especially problematic for the offshore and constant seawater treatments where we were simulating these conditions with temperature-controlled units. Instead, we created a reservoir tank where we could very accurately control these conditions for each treatment more effectively. Therefore, we attempted to minimize a lack of independence but are confident that keeping females isolated within the same tanks was acceptable given the research question and associated analyses we sought to explore.

At each sampling interval, lobster eggs ($\sim$100/sample) were removed from the center of each lobster clutch with a pair of fine forceps. All egg samples were rinsed and gently agitated with a 0.5% sodium hypochlorite and distilled water solution for $\sim$1 min., after which they were rinsed with 100% distilled water and blotted dry to remove the cement matrix holding the eggs together (P. Talbot, personal communication, 2011). Previous studies indicated that this chemical separation technique was non-invasive and did not compromise the biochemical integrity of the eggs due to their complex and thickened membranes (*Johnson, Goldstein & Watson III, 2011*). Each egg sample was divided into 30 egg aliquots then freeze-dried at $-40\,^{\circ}$C for 24 hr. (Labconco Freeze Dryer 5, Kansas City, MO) and served as a stock for subsequent analyses. Samples were then ground down into a fine powder using an industrial-grade milling machine (Wiley Mill #4, 40 $\mu$m mesh screen, Thomas Scientific, Swedesboro, NJ) and samples were stored in labeled polyethelene scintillation storage vials for subsequent analyses.

## Biochemical analyses

For each of the five sampling intervals, we determined total lipids and proteins for each lobster in triplicate ($n = 30 \times 3$ egg samples/female) using the methods described in *Goldstein (2012)*. In brief, total protein levels were determined using a modified Lowry method (*Lowry et al., 1951*) using a BioRad protein assay kit with Coomassie Brilliant Blue G-250 (reagent standard) and bovine serum albumin as a standard (Biorad Laboratories, Hercules, CA). Egg samples were digested in 1N NaOH, filtered, and read on a spectrophotometer (Beckman DU-250; $\lambda = 595$). Total lipid was quantified gravimetrically using the general protocol detailed in *Bligh & Dyer (1959)*. The procedure was modified to use a ratio of 1:2:2.5 chloroform-methanol-water extraction, respectively. Samples were dried for 24 hr at 37 $^{\circ}$C and stored in a glass desiccator, before being weighed on an analytic balance.

## Egg volumes

For calculating egg volumes, 10-15 fresh eggs were removed in the middle of each of four sampling months (Sep, Nov, Jan, and Feb) from each lobster and placed in plastic 2.0 mL storage tubes, preserved in a 4% formalin and sterile seawater solution and stored at 4 °C. Although we had intended to sample and calculate egg volumes in accordance with the same temporal sampling regimes as our biochemical assays, for logistical reasons we were not able to do so. For each egg, a digital picture was taken using a Nikon Coolpix 995 digital camera mounted to a dissecting microsope (Nikon SMZ-2T, Nikon USA Inc., Melville, NY). All egg images were imported into an image processing software (Image J v.1.35, see http://rsb.info.nih.gov/ij/) and a digital measuring tool was used to make calculations of each egg's longest two axes. All egg calculations were measured to the nearest 0.1 mm (then converted to $\mu$m) and values for each sample were averaged ($\pm$se). Egg volumes were then calculated using the formula: $V = 4/3 * (\pi r^3)$, where $r$ is the radius for spheroid-shaped embryos (see *Garcia-Guerrero & Hendrickx, 2004*).

## Data analysis

Analysis of variance (ANOVA) was used to investigate potential differences in egg protein and lipid content between the three thermal regimes (fixed factor 1) at each of the five sampling intervals (fixed factor 2). A $3 \times 5$ full factorial design was used and analyzed as a split-plot (SP) ANOVA (whole-plot = temperature, sub-plot = month, $df_{total} = 15$) using a PROC MIXED model in SAS v. 9.3 (SAS Institute Inc., Cary, NC). For all ANOVA models, we tested the assumptions of normality, independence, and homogeneity of variance and used a Kuskal Wallis $H$-test where these assumptions were not supported. Differences between groups were compared using the PDIFF function in SAS. Regression analyses were carried out using JMP v. 9.3 (SAS Institute Inc., Cary, NC) statistical software. All means are expressed $\pm$ se.

# RESULTS

## Water temperatures

Seawater temperatures over the entire course of this study (October-May) averaged $7.1 \pm 0.24$ °C (range = 2.1–11.2) for inshore laboratory trials, compared with $6.0 \pm 0.19$ °C (range = 2.8–10.1) for the offshore thermal regime, and $16.2 \pm 0.21$ °C for the constant treatment tank (Fig. 1). There was an overall significant difference in water temperatures between the constant tank treatment and both inshore and offshore ones (*ANOVA, Kruskal-Wallis H-Test*; $F_{2,7} = 10.32$, $P < 0.0001$; Fig. 1) but not between inshore and offshore. It is worth noting that temperatures in the inshore and offshore treatments converged quickly at the outset of the experiment and diverged markedly starting only in early April (Fig. 1). Thus, it is not surprising that there was no difference in embryo development between inshore and offshore treatments, at least until the March sampling, however inshore temperatures were warmer in April and May compared with offshore locations ($P < 0.05$).

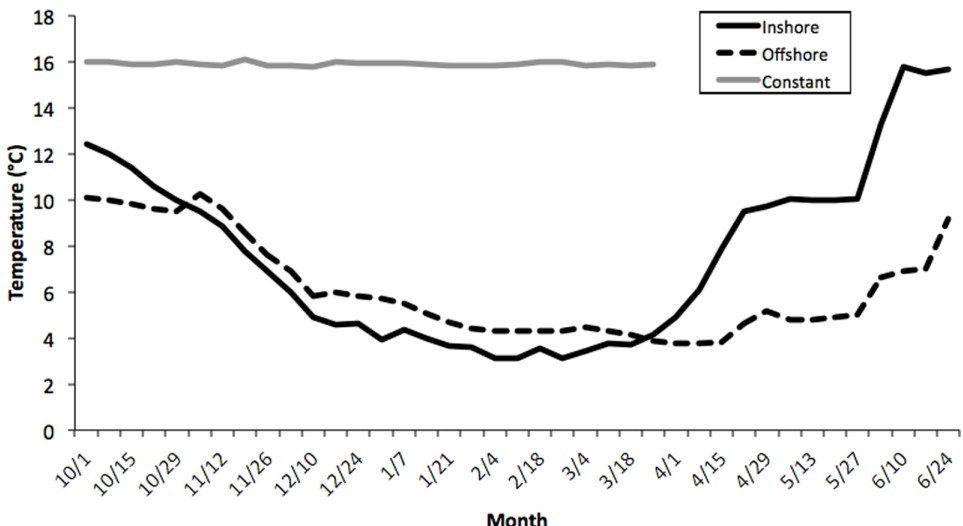

**Figure 1 Average temperature profiles for three thermal treatments.** Weekly temperature averages compiled for three thermal treatments: inshore, offshore, and constant from October–June, 2006–2007. See Methods section for details. There was an overall significant difference in water temperatures between the constant tank treatment and both inshore and offshore ones (*ANOVA, Kruskal–Wallis H-Test*; $F_{2,7} =$ 10.32, $P < 0.0001$) but not between inshore and offshore.

## Lipid and protein content

Both total egg lipid and protein levels from inshore and offshore thermal regimes differed from their constant temperature counterpart (*SPANOVA; P = 0.0002*, Fig. 2). Likewise, the interactive effect of temperature and month was significant for both lipid ($F_{7,44} = 2.27$, $P < 0.045$) and protein levels ($F_{7,44} = 46.5$, $P < 0.0001$, Table 1). Overall egg lipid values showed a decrease over time (equation: lipids $= 381.76 - 55.00^*$month, $r^2_{adj} = 0.85$, $P < 0.0001$; Fig. 2), falling most dramatically early in the fall ($-16.8\%$ inshore, $-21.4\%$ offshore, $-24.8\%$ constant) and late spring ($-63.7\%$ inshore, $-59.0\%$ offshore). By contrast, total lobster egg protein values increased over the same time-frame (equation: proteins $= -35.53 + 69.11^*$month, $r^2_{adj} = 0.63$, $P < 0.0001$; Fig. 2), and exhibited the largest increases in the fall (60.4% inshore, 57.7% offshore, 66.5% constant) and spring (30.1% inshore, 37.1% offshore).

## Egg volumes

Overall, there was a significant increase in egg volume over time for all eggs over all treatments ($r^2_{adj} = 0.413$, $P < 0.001$). However, there were no significant changes with respect to egg volume by treatment ($F = 0.73$, $df = 2$, $P = 0.513$) (overall means: inshore $= 3{,}226 \pm 163$ µm$^3$, offshore $= 3{,}254 \pm 167$ µm$^3$, constant $= 3{,}476 \pm 152$ µm$^3$), even though there were differences from month-to-month ($F = 2.25$, $df = 3$, $P < 0.001$; Fig. 3).

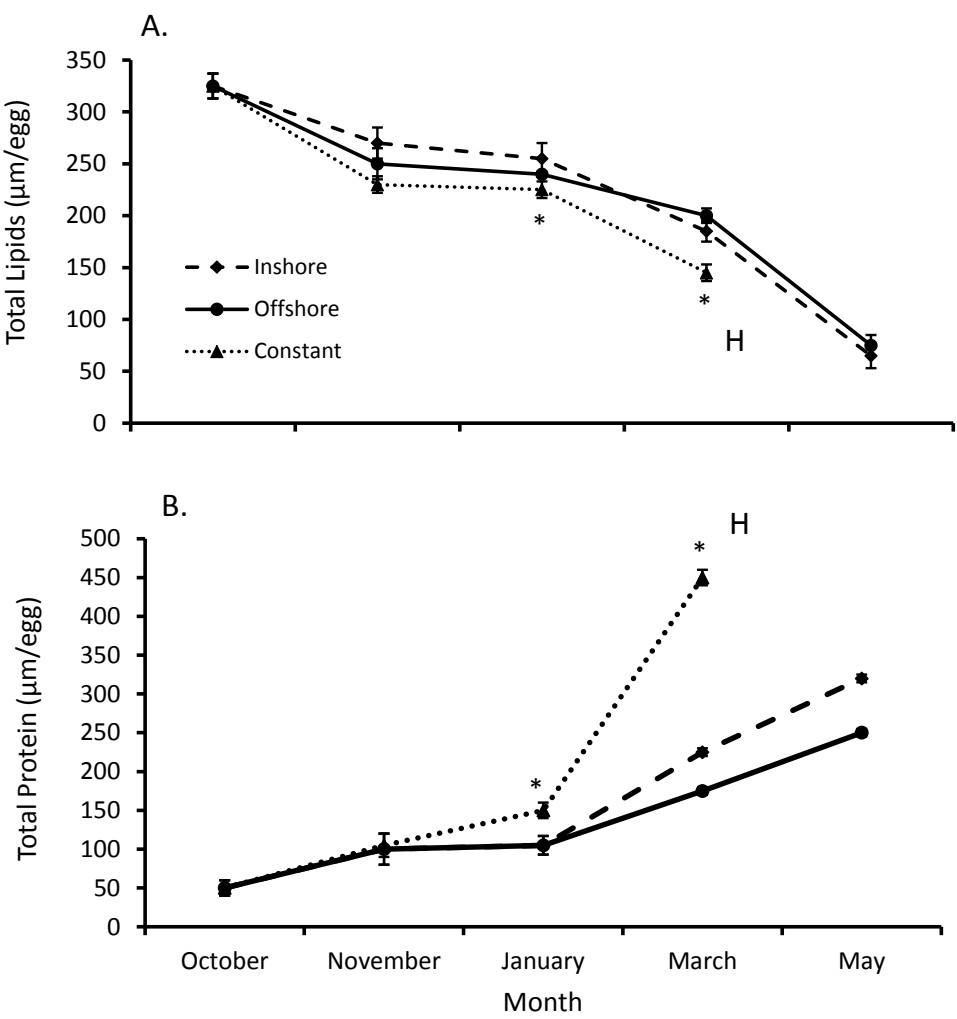

**Figure 2 Changes in lobster egg lipids and proteins.** (A) Change in lipids and (B) protein levels (means ± se) over the course of seven months of egg development for all lobsters sampled (*n* = 5/trt). Posthoc differences (*P* < 0.001) between treatments denoted with an *. Lobsters exposed to inshore and offshore thermal treatments did not hatch their eggs until after May, unlike eggs from the constant treatment, where eggs hatched (H) in April.

**Table 1 Lipid and protein values with temperature.** Pairwise comparisons (*p*-values) between temperature treatment and month for both lipids and protein values. Boldface *P*-values (< 0.05) denote significant differences between temperatures for a specific month.

| Treatment | October | November | January | March | May |
|---|---|---|---|---|---|
| Inshore * offshore | 0.85 | 0.30 | 0.21 | 0.25 | 0.24 |
| Inshore * constant | 0.89 | **0.03** | **0.002** | **0.002** | – |
| Constant * offshore | 0.72 | 0.22 | **0.04** | **<0.0001** | – |

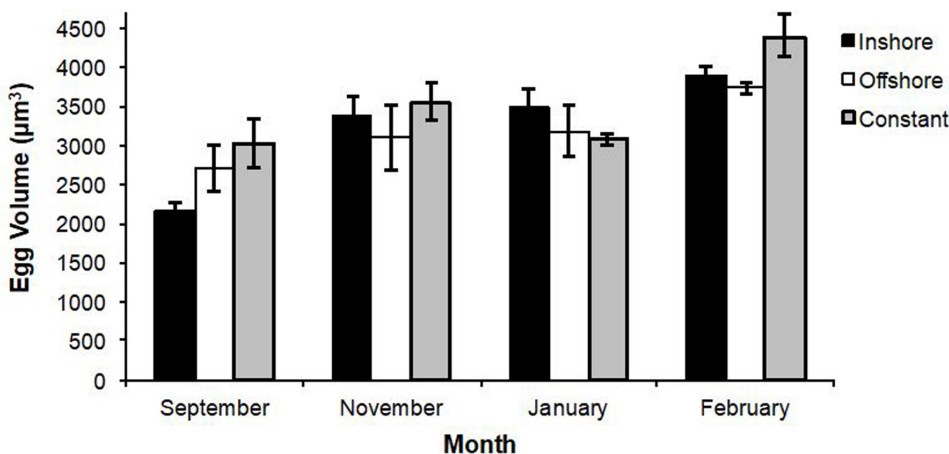

**Figure 3  Changes in lobster egg volumes.** A summary of means (±se) for changes in lobster egg volumes (given in μm³) over a six month period. There were no significant differences in egg volume by treatment (Tukey's HSD; $q = 2.40$, $P > 0.05$), but differences did exist from month-to-month ($F = 2.25$, $df = 3$, $P < 0.001$).

# DISCUSSION

Our main goal was to document the changes in lipids and proteins in lobster eggs as they developed during exposure to three different thermal regimes in the laboratory. In general, we found that the trends during embryogenesis in *H. americanus* were typical of other decapods: lipid reserves were catabolized while proteins were utilized to make tissues (*Holland, 1978*; *Sasaki, McDowell Capuzzo & Biesiot, 1986*; *Jacobs et al., 2003*; *Brillon, Lambert & Dodson, 2005*). In tandem with these patterns, eggs were also shown to increase in diameter. Not surprisingly, lobster eggs exposed to an elevated, constant temperature developed faster and thus used up more of their energy reserves (lipids) sooner than eggs subjected to natural thermal regimes, which included decreases in water temperatures from November to March.

In this study we did not obtain data for biochemical changes that occurred in eggs that were approaching hatch (~30 days prior), nor the effects of the different thermal regimes on larval survivorship or condition but a complementary study did find differences in time of hatch (*Goldstein & Watson III, 2015a*). This may have been why there were no apparent biochemical differences in lobster eggs between inshore and offshore temperature treatments, even though inshore temperatures increased more rapidly in the spring. Previous studies have shown that large changes in egg yolk lipids and protein levels occur within the last few weeks of development (*Sibert, Ouellet & Brethes, 2004*), and we reported that eggs exposed to inshore thermal regimes hatch sooner than those that experience offshore conditions (*Goldstein & Watson III, 2015a*; *Goldstein & Watson III, 2015b*). Therefore, we expect there would have been differences in egg biochemistry if we had been able to obtain samples just before hatching in the spring.

Other studies have shown the influence of varying thermal exposures on larval condition (*Sasaki, McDowell Capuzzo & Biesiot, 1986*; *Ouellet & Plante, 2004*), and it was very
clear that significant changes to lobster egg biochemistry are apparent in the first two months of development (this study) as well as leading up to the month before hatching (*Sasaki, McDowell Capuzzo & Biesiot, 1986*). The effect of temperature on metabolic and developmental rates is expressed through changes in the consumption rates of metabolic reserves that are affected by changing temperatures (*Sasaki, McDowell Capuzzo & Biesiot, 1986*). Thus, the seasonal aspects of fluctuating temperature impact the rates and course of development in lobster eggs. It is suggested that fluctuating seasonal temperatures help to accelerate egg development during some time frames while depressing it at others, providing temporal windows where hatching generally takes place (*Templeman, 1940*; *Helluy & Beltz, 1991*; *Waddy & Aiken, 1995*; *Goldstein & Watson III, 2015b*).

Seasonal movements by ovigerous lobsters may have evolved in order to expose their eggs to the most suitable available water temperatures for development, or to ensure that females reside in ideal locations when these eggs hatch to increase larval survival (*Gendron & Ouellet, 2009*; *Goldstein & Watson III, 2015b*). This certainly seems to be the case in movements of late-stage ovigerous Caribbean spiny lobster (*Panulirus argus*) where these animals make homing excursions from their dens on the reef to the reef edge to release their larvae (*Bertelsen & Hornbeck, 2009*). It is also possible that, by moving offshore, egg incubation time is actually slowed down, which might both conserve internal egg resources and delay hatch until environmental conditions are optimal in the spring. Seasonally changing temperatures, including a refractory period of cold seawater temperatures (<5 °C) that elicits a stasis in eggs are important to conserving egg resources for more rapid increases in temperature (>10 °C) that typically occur later on (*Waddy & Aiken, 1995*).

## Lipids and proteins

Many studies conducted on crustacean eggs show that lipids are the major energy reserve (*Holland, 1978*; *Fraser, 1989*; *Clarke, Brown & Holmes, 1990*; *Heras, Gonzales-Baro & Pollero, 2000*). Although egg yolk lipids were consumed at disparate rates in our thermal treatments and throughout all months, lipids were consumed much more modestly in winter (Fig. 2). This pattern is seen consistently in other crustaceans as well. For example, the egg lipid content of fiddler crab (*Uca rapax)* decreases significantly (78.4%) through embryogenesis, confirming that lipids constitute an important energy source for embryonic development (*Figueiredo et al., 2008*). In addition, lipids are also used as structural components of cell membranes that are being formed as they grow (*Rosa & Nunes, 2003*). Thus, the catabolism of lipids is a classic feature of crustacean eggs and many other crustaceans produce eggs with large lipid reserves that are used throughout embryogenesis (*Rosa et al., 2007*). Lipid depletion rates are directly related to incubation temperature, and it has been observed in other crustaceans that the energy consumption per day slightly intensified 3 or 4 days before hatching, which could be related to a higher energy production need at this time (*Heras, Gonzales-Baro & Pollero, 2000*). Yolk lipids tend to become catabolized first followed by yolk proteins. These ratios change and can be used to estimate the cost of egg development at differing temperatures (*Sasaki, McDowell Capuzzo & Biesiot, 1986*). In the field, lipid profiles (e.g., fatty acids) have been used to identify offshore from inshore lobster eggs (*Castell et al., 1995*); therefore, it is possible that these

constituents are utilized differently across different geographic regions that correspond to contrasting thermal regimes.

For proteins, the consumption rate during embryogenesis may increase as temperature rises (*Conceicao et al., 1998*). Proteins both function as building blocks for tissues and as a source of energy, when needed (*Schmidt-Nielsen, 1991*). At elevated temperatures (constant), increases in protein levels were detected and, at these elevated temperatures, tissue synthesis tends to be inefficient and more protein might be used for energy instead (e.g., *Garcia-Guerrero, Racotta & Villareal, 2003*; *García-Echauri & Jeffs, 2018*). Therefore, the duration and rates of differing thermal profiles would most certainly affect these biochemical changes and allocations of resource components over time. How this translates to larval survivorship remains poorly understood. However, *Sasaki, McDowell Capuzzo & Biesiot (1986)* showed that up until Stage IV (*i.e.,* transitionary postlarval stage), lobsters depended, in-part, upon stored lipids and that proper temperature synchronization in these residual lipids maybe favorable to settlement processes.

## Egg volume

Increases in egg volume are primarily due to water uptake by the embryo as well as from the retention of metabolic water resulting from respiration (*Pandian, 1970*; *Petersen & Anger, 1997*). The associated osmotic changes that ensue during egg development can be an important component to hatching and have also been implicated in mechanically aiding the breakage of the chorion near the time of hatch (*Pandian, 1970*). Slight changes in lobster egg volume have been previously explained as a function of a plastic response to variations in salinity (*Charmantier & Aiken, 1987*), and for later eggs, a consequence of physiological factors during development (*Pinheiro & Hattori, 2003*). In these instances, the movements or residency of lobsters in certain locations where seawater salinities can vary dramatically during certain times of the year (e.g., estuaries; *Watson III, Vetrovs & Howell, 1999*) may have an impact on aspects of development or hatch, especially near the latter part of egg development (*Charmantier & Aiken, 1987*).

## Female size and condition

In this study we did not specifically address the influence of maternal size or nutritional condition on egg quality in *H. americanus*. However, other related studies have showed that caloric energy content per egg increases with female size (*Attard & Hudon, 1987*). *Sibert, Ouellet & Brethes (2004)* described this relationship by creating a growth index model for egg development and found that larger eggs used yolk lipids more efficiently and sustained faster embryonic growth compared with smaller eggs. An effect of female size on egg reserve allocation has been reported in other decapods including snow crab (*Chionoecetes opilio*), giant crab (*Pseudocarcius gigas*), and lobster (*Homarus americanus*) (*Attard & Hudon, 1987*; *Sainte-Marie, 1993*; *Gardner, 2001*). In lobsters it has been postulated that the effect of female size may mean that larger females make a greater contribution towards egg reserves (*Attard & Hudon, 1987*). However, the added effect of temperature on egg quality may, in some cases override this effect and maternal nutrition may also modulate egg quality (*Goldstein & Shields, 2018*). The lecithotrophic nature of lobster eggs is determined largely

through the sequestering of maternal nutrients throughout the processes of primary and secondary vitellogenesis during oocyte formation, the latter of which is highly dependent on the female's organic energy reserves (e.g., lipoprotein; *Dehn, Aiken & Waddy, 1983*).

## CONCLUSIONS

Although the changes in biochemical components (lipids and proteins) in developing lobster eggs were not significantly different from inshore and offshore thermal regimes, the potential exits for variations in the energetics of embryogenesis influenced by the seasonal movements of some lobsters to and from these two disparate locations over the entire course of egg development from egg extrusion to hatch. Because we did not see differences, in egg biochemistry, inshore eggs hatched sooner even though the protein and lipid levels were the same as offshore eggs. As seasonal thermal cycles fluctuate or potentially shift (i.e., climate change), the timing of egg hatch and associated egg quality may modulate biochemical changes to lobster eggs and have implications for hatch, larval energetics, and ultimately, hatch-to-recruitment dynamics for this important fishery.

## ACKNOWLEDGEMENTS

A special thanks to Nancy Whitehouse of the UNH Dairy Nutritional Research Center whose expertise and advice on biochemical analyses and statistics were very helpful to the completion of this project in addition to the use and training of diagnostic lab equipment. Project interns, Sarah Havener, May Grose, and Michelle Provencier, were also very helpful with many aspects of this study and their help is much appreciated. Nate Rennels and Noel Carlson of the UNH Coastal Marine Lab for their help in maintaining lobsters and the seawater system. A special appreciation to NH lobstermen Alan Vangile (F/V Special K) and Michael Pawluk (F/V Gretchen D) for the boat hours to trap and collect egg-bearing lobsters for this study. We also thank K. Liversage and one anonymous reviewer for their constructive recommendations for improving this manuscript.

### Funding

Funding support for this project was provided by a NH Sea Grant (project # R/MED-9) to Winsor H. Watson and Jason S. Goldstein, a Northeast Consortium Grant (NOAA Grant # 111856) to Winsor H. Watson and a UNH Marine Program Grant to Jason S. Goldstein. The funders had no role in study design, data collection and analysis, decision to publish, or preparation of the manuscript.

### Grant Disclosures

The following grant information was disclosed by the authors:
NH Sea Grant (project # R/MED-9).
Northeast Consortium Grant: NOAA Grant # 111856.
UNH Marine Program.

## Competing Interests

The authors declare there are no competing interests.

## Author Contributions

- Jason S. Goldstein conceived and designed the experiments, performed the experiments, analyzed the data, prepared figures and/or tables, authored or reviewed drafts of the paper, approved the final draft.
- Winsor H. Watson III contributed reagents/materials/analysis tools, authored or reviewed drafts of the paper, approved the final draft, experimental design advice and data collection.

## Field Study Permissions

The following information was supplied relating to field study approvals (i.e., approving body and any reference numbers):

This work was legally permitted under the State of New Hampshire Department of Fish and Game (Permit RSA 214:29).

## Data Availability

Raw data is available in a Supplemental File.

## Supplemental Information

Supplemental information for this article can be found online at http://dx.doi.org/10.7717/peerj.6952#supplemental-information.

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
