# Peer review of "Biochemical changes throughout early- and middle-stages of embryogenesis in lobsters (Homarus americanus) under different thermal regimes"

_PeerJ, doi:10.7717/peerj.6952_

## Round 0.1 · original submission · Major Revisions

Both reviewers have made extensive comments and suggestions to improve the paper. I think that if you address their comments (and see my annotated pdf) this could be a good paper. You can certainly shorten it a bit by avoiding repeating data in both tables and figures (perhaps submit the data in the table as supplementary?

Reviewer 1 ·

Basic reporting

Please see attached pdf file.
English needs improvement.
Article structure ok but Figures 1 and 2 not needed.
Discussion should be more focused and stick to actual results.

Experimental design

Please see attached pdf file.
In general, the methods should be better explained.
Not sure egg volumes and biochemical content were measured at the same times, although methods suggest this was the case.
Measurements of egg biochemical content did not extend to close to time of hatching in the "inshore" and "offshore" treatments, compromising strong conclusions about the adaptive value for embryo development of thermoregulation through active movement of ovigerous females.

Validity of the findings

Please see attached pdf file.
The data appear to be sound, although some information is unspecified.

Annotated reviews are not available for download in order to protect the identity of reviewers who chose to remain anonymous.

·

Basic reporting

The basic reporting is well done. The writing style was a pleasure to read and the authors have done a good job with referencing the related background literature.

I will only make two points concerning the article structure:

1) I do not think the first two figures are really needed as they just show some photos of lab equipment.

2) I do not see why figure 4 shouldn't be incorporated as a line graph into figure 3, as it has the same predictor variables.

Experimental design

I think there are some issues with the experimental design:

1) The main issue is pseudoreplication. According to Hurlbert's review on pseudoreplication, the setup used in this experiment is classified as "isolative segregation" which is stated as one of the "various ways in which the principle of interspersion can be violated" (see Figure 1, Hurlbert, S.H., 1984. Pseudoreplication and the design of ecological field experiments. Ecological monographs, 54: 187-211). Ideally each lobster would be housed in is own separate tank and thus the eggs would be independent replicates that are randomly interspersed among other replicates. I think the authors will need to argue how their experimental setup can be valid despite the lack of interspersion of their treatments. Potentially the authors may be able to demonstrate that conditions (besides temperature) were similar among the treatments despite the fact they were isolated.

2) I had trouble following some aspects of the methods descriptions. Line 155-156 states there were 2 tanks used per treatment, and there were 3 treatments, so I was expecting 6 tanks, but line 165 states there were 4 tanks. Also, having 4 tanks does not seem to fit with having a total of 15 lobsters. Thus I was not able to work out how the experimental setup was organised, although it might become clear if the methods writing is changed/expanded and/or a diagram provided.

3) For this experimental design to work, all the treatments should have had eggs with the same level of development at the start of the experiment. From figure 3 it does seem the lipids and proteins were very similar among treatments, but in Figure 5 it seems there may have been differences in size at the 1st month. I would ask that the authors check (e.g. ANOVA) that all egg variables were statistically similar among treatments when the experiment started.

4) ANOVA assumes homogeneity of data variance, which does not appear to have been tested (e.g. Cochran's test or Levene's test). It could also be argued that as only 15 lobster were used, the sample size is quite low in which case normality of distributions will also be an important assumption to test. I would ask that the authors please address these aspects of their statistics.

5) On lines 153-155 there is a brief comment that "A subset of the eggs in each clutch were viewed under a dissecting scope and staged... These samples also served as covariates for all subsequent statistical analyses". The use of covariates is never mentioned again in the Data Analysis section nor any results concerning covariates in the results section. If covariates are used then the analysis will be an ANCOVA (not ANOVA as stated on line 223). If the analyses really did have a covariate then I would ask that there is more description of why and how this was done.

6) In the lipid and protein analyses, the constant temperature treatment was missing the last data point because the eggs hatched, and thus the design is unbalanced (i.e. May data for only 2 out of 3 treatments); was the statistical procedure used able to handle an unbalanced design? It would be my preference to have finalized the whole experiment as soon as it becomes impossible to continue getting data from one of the treatments. Overall, the fact that the design was unbalanced needs to be discussed.

7) I would say that the regression analyses (Figure 4) are not needed, as the significant effect of time and the pairwise tests in the ANOVA analyses provide all this information.

Validity of the findings

I would say that the validity of the findings is not certain at this stage due to potential issues with pseudoreplication and analysis assumptions, and the authors will need to do more convincing for the reader that their conclusions are valid.

Additional comments

Overall, if efforts were made to present this study in a concise way then it could have only 1 figure of results and perhaps 1 table. The discussion is well written and full of background information, but much of this is loosely related to the topic at hand and if this were reduced then the manuscript could probably only be considered as a short communication. Although it may be possible to work through most of the issues about experimental design, given that there are quite a few of them, and that the volume of results is quite low, I would be hesitant to support this manuscript without substantial and possibly fundamental changes.

here are a few further minor comments:

1) It would be very useful to see statistics tables and these will help the reader work out how the experiment was set up and the analyses done.

2) Line 221: Here you mention sampling of egg volumes was done at 5 times but Fig. 5 shows only 4 sampling times. Also on Fig. 5, why are the months different to the sampling months on the other graphs?

3) Please put SE bars on all data graphs.

4) I think all ecological laboratory studies should have at least some discussion/mention about the limitations of this approach. With enough effort it would be possible to measure lobster egg development in the field which would provide much higher quality results.

---

## Round 0.2 · Minor Revisions

Both reviewers have some further useful comments on your submission. I suggest you remove the stats from your abstract they are not really necessary there.

Reviewer 1 ·

Basic reporting

Please see attached review.

Experimental design

Please see attached review.

Validity of the findings

Please see attached review.

Additional comments

Please see attached review.

Annotated reviews are not available for download in order to protect the identity of reviewers who chose to remain anonymous.

·

Basic reporting

I have some suggestions to improve the basic reporting:

Line 58, 59: words “both” and “alike” maybe not needed
Line 62: change “as” to “are”
Line 66: remove word “also”
Line 88: I would say the word “lab” may be too colloquial
Line 102: change “we sought to test” to “we tested”
Line 150: maybe to increase succinctness remove “and placed in labeled plastic sample trays”.
Line 211: maybe remove “very”
Figure 2: please add standard error bars to the sampling points on the line graphs
Line 217: change “and these results are summarized in Tables 1 & 2” to “(Tables 1 & 2)”
Table 1: the means in this table appear to be exactly duplicating the information on Fig 2. I think the standard errors and post-hoc tests should be presented on Fig. 2 and Table 1 removed.
Table 2 legend: please mention in the 1st sentence that the values shown are P values.
Line 236: change “The main goal of this study was” to “Our main goal was”
Line 267: change “couple” to “two”
Line 272: The use of the word “real” here is a bit confusing, I would revise.
Line 305-309: this long sentence could be split in two.
Line 384: Change “dramatically” to “significantly”
Figure 2, 3: It is confusing for the reader that bar graphs were used Fig. 3 and line graphs Fig. 2, when they both show very similar time series. I would recommend for consistency to change all time series as line graphs, unless there is some specific reason its like this.

Experimental design

I think it would be useful for some of the discussion about pseudo-replication in the response to reviewers document to be briefly mentioned somewhere in the manuscript.

Validity of the findings

Line 102-104: The hypotheses should directly match the tests done (i.e. the actual statistical comparisons). The only factors tested here were temperature and time; no measurements of migration were made. Also, the temperatures were not changed according to any migrations, they just tested how eggs would develop from lobsters staying either inshore or offshore. So the hypotheses should state something like “It was hypothesized that different water temperatures, simulated in the laboratory to be similar to temperatures experienced by lobsters in varying coastal zones (inshore vs offshore), will affect egg biochemistry. This hypothesis was tested to gain information useful for understanding how egg development may be affected by lobsters migrating between those zones”

Line 207: the lack of significant difference in temperature between inshore and offshore water was surprising and may influence the validity of the conclusions, as this study is meant to inform about lobsters migrating between these areas to expose their eggs to different temperatures. Please provide some text about the implications of the non-significant temperature result in the discussion.

Line 324-326: I think this text needs to be clarified/corrected. I guess the “constant” temperature being referred to here is the one used in this experiment, which according to line 248 appears more optimal for growth, but here it is stated as “sub-optimal”. Also, if more protein is used for energy as stated here, then there should be less rather than more protein in the “constant” treatment which is not the case (Fig. 2).

To allow the reader to understand the conclusions better, please somewhere early in the discussion provide a clear statement about whether or not the results supported the specific hypotheses from the introduction.

Line 384-385: Please mention here why “there is still the potential”, rather than just saying that the potential exists. It seems the results may indicate that no evidence could be found to support the hypothesis that “variations in the energetics of embryogenesis influenced by the seasonal movements of some lobsters to and from these two disparate locations”, especially since these disparate locations did not even have significantly different temperatures according to my understanding (line 207). Also I think the word “are” is needed between “embryogenesis” and “influenced”.

Additional comments

I think a concluding sentence in the abstract stating the relevance of the results would be useful.

Line 39: add “potentially” between “thereby” and “exposing”, especially since this study did not appear to find any significant different (Line 207) between the two areas where migrations occur between.

Line 130: Here it says the “constant” treatment was 12 degrees, but in figure 1 it seems this treatment had 16 degrees. Please correct one or the other.

Line 211-217: According to how I understood this, here all the results are from the 2 factor analyses. Firstly the text here details the main effects, then results of the interactions. The interactions are all significant, which makes it redundant to mention the main effects as was done first (i.e. effects of the factors in isolation are not informative). If this understanding is correct, then the text from line 211-215 may not be needed.

Line 300-302: a reference needs to be provided for this statement.

Line 350-378: These paragraphs are on a different topic to the study and are sufficiently unrelated that they confuse the story a bit and I would say they are probably not needed, or could be reduced to a few sentences.

---

## Author Rebuttal · Round 0.2

## Wells National Estuarine Research Reserve

Research • Education • Stewardship

wellsreserve

October 30, 2018

PeerJ, Inc.
PO Box 910224
San Diego, CA 92191
email: authors@peerj.com

Dear PeerJ editorial staff,

We are pleased to resubmit to you and your editorial staff our revised manuscript (Article ID 25864), *'Biochemical changes throughout early- and middle-stages of embryogenesis in lobsters (Homarus americanus) under varying thermal regimes'*, for re-assessment and hope that you and the two Reviewers find our improvements acceptable for approval.

We received two reviews, both positive and reasonable, that recognized the strengths and merits of this study regarding our current knowledge lobster egg development and some of the biochemical properties associated with that process over a suite of differing thermal regimes. The first Reviewer placed an emphasis on improving the writing and clarifying our methods, results, and discussion towards a more focused approach. The second Reviewer was pleased with our writing and reporting but did have concerns regarding our experimental design, statistics, and the most efficient use of figures in the manuscript. This Reviewer also had several good suggestions where we needed to revise our text.

Since this time (and we apologize for the extended response), we have been able to fully address both Reviewers' important comments and suggestions, and have corrected, clarified, or improved just about all these issues that are detailed in a point-by-point response. For example, we have eliminated two Figures that were not too informative and added in one new Figure that was suggested. We have fixed and/or clarified ambiguous writing, and qualified some of our statements as well. We are providing a marked-up version of the manuscript and a 'clean' version so that all changes that were made can be seen.

We hope that this submission will be accepted and become a contribution to PeerJ. If you have any questions, please do not hesitate to contact me.

With Kind Regards,

Jason S. Goldstein, PhD.

**PeerJ Article ID: 25864**
**Reviewer Comments and Responses**

Article Title: Review of "Biochemical changes throughout early- and middle-stages of embryogenesis in lobsters (*Homarus americanus*) under varying thermal regimes" by J.S. Goldstein and W.H. Watson III, submitted to PeerJ.

## Reviewer-1

General comment

The biochemical analyses appear to have been done carefully and are rather interesting. The writing needs to be checked: some paragraphs and sentences are poorly structured (see for example major comment no. 1, 3 and all minor comments below). The description of methods and statistical analyses should be improved (see major comments no. 2, 4, 5, 6, 9). The discussion could be better focused and written more concisely.

Major comments

1. Lines 90-95. This is a long, very awkward sentence. Break it into two sentences (description of stages and effect of temperature on duration of the egg/embryo stage). Also, as currently formulated, it appears that embryo and egg are two different developmental stages. The embryo is in the egg.

*Response: This has been revised (LINES 209-215).*

2. Line 153. "A subset of the eggs in each clutch". How many eggs were in that subset? And when was this preliminary screening done (give exact dates)?

*Response: We examined a subset of 15 eggs/female on Sept-5, and we have included this information on LINES 291-292.*

3. Lines 153-157. Describe the protocol for measuring and tagging ovigerous female lobsters (lines 157-160) before describing egg sampling and staging (lines 153-157). The size range of female lobsters used in this experiment should be reported.

*Response: We agree – this makes sense logically. We have moved this section (LINE 288-).*

4. Lines 153-157. "These samples also served as covariates for all subsequent analyses". It is unclear what, other than embryo stage, was measured on this initial sample of eggs (volume, weight, biochemical content) and used as a covariate? The values obtained for these initial samples do not appear to be reported in the results section. And a covariate or covariates are not included in the model description of the "Data analysis" subsection (lines 223-227). Please clarify.

*Response: This is quite right and we never really used the data in this manner. We have removed this statement in the text.*

5. Line 178. Were the 5 ovigerous females composing this subset the same all through the 5 sampling periods?

*Response: Yes, we kept the same females throughout. We clarified this a bit (LINES 328-329).*

6. Lines 179-180. "All lobsters were sampled at five discrete time periods: twice in the fall and spring… and once in the winter." All lobsters, or the subset of 5 ovigerous females. Also, specify the dates for sampling (day and month).

*Response: All lobsters were sampled, 5 lobsters for each treatment for a total of 15. This statement and the specification of sampling dates have been added in LINES 134-138.*

7. Line 199. These three samples each contained 30 eggs, correct?

*Response: Yes, that's correct and this has been clarified in LINES 148, 155 & 163.*

8. Lines 234-235. It would probably be useful to shown the temporal variability in temperature for the inshore and offshore treatments, in a separate graph or an overlay on Fig. 3 with a new right-hand Y-axis for reference.

*Response: We agree and have produced a temperature graph for each of our three thermal treatments (inshore, offshore, and constant) and have substituted this as a new Figure. 1 that complements the first part of the Results section.*

9. Lines 251-256 (in reference to data in Figs 3 and 4) and lines 264-266 (in reference to Fig. 5). Methods state on lines 179-180 that "All lobsters were sampled [for biochemical analyses] at five discrete time periods: twice in the fall and spring… and once in the winter" and on line 211 that "For calculating volumes, 10-15 eggs were removed at each of the aforementioned five time periods". So if lobster eggs were sampled at the same five times for biochemical analyses and egg volume determinations, why is it the case that there are 5 temporal observations of biochemical condition in Figs 3 and 4, for the months of October, November, January, March and May, but only 4 temporal observations of egg volume in Fig. 5 of which two (September and February) are in different months than those for biochemical condition?

*Response: This is good thing to point out and we did (unfortunately) sample eggs for volume at offset temporal intervals. We did this because we did not initially intend to examine egg volume but decided to do so after egg sampling for biochemical assays had already started. Additionally, we had no intentions to try and match up egg volume with biochemical data. Still, the four sampling intervals that we did choose (Oct, Nov, Jan, Mar, May) do encompass a temporal sampling window reflecting a full season of growth, which one could argue represent a full season of growth. Therefore, we have attempted to clarify this in the manuscript (LINES 174-179).*

Where are the October, March and May observations for egg volume? Furthermore, Figs 3 and 4 are made up as if sampling occurred on the 15th of each month – is this really the case?

*Response: Please see comments above.*

10. Lines 274 and 364. Although egg volume increased over time in the study, I see no direct evidence (data on water content) supporting the statement that "eggs were also shown to absorb water" or "The increase of water in the eggs (egg volume) as seen in this study".

*Response: Although other studies have shown that crustacean eggs increase in size to water absorbance, we did not directly test this and have removed this claim in the text (LINE 241).*

11. Line 327. "Egg yolks were rapidly consumed in all thermal treatments". This seems to contradict other statements in the discussion, e.g. lines 320-321. Indeed, the relative magnitude of yolk reduction

was much greater in the inshore and offshore treatments than in the constant temperature treatment (Fig. 3). As a consequence, larvae hatching from this last treatment would still have substantial yolk reserves and could thus possibly be sustained for some time without external feeding (see lines 313-315). This consideration is theoretical though, as constant thermal regimes do not exist within the lobster's natural range.

*Response: We agree that this is not as clear as it could be and have made changes to the text as such (LINES 294-296).*

12. Lines 354-355. Are the authors declaring the inshore and offshore thermal regimes to be suboptimal? Increases in protein content also occurred in these two thermal regimes, although they reached smaller levels at a slower rate than in the constant temperature treatment.

*Response: This was certainly not the intention and thank you for catching that inaccuracy. We have gone ahead and changed that too (LINES 322-323).*

13. Figures 1 and 2 are not very useful.

*Response: We have eliminated these two Figures and have re-numbered the remaining Figures accordingly (but see comment 8 above).*

Minor comments

Line 3. Title: change "varying" to "different"… one of the experimental thermal regimes is a constant temperature.

*Response: This does make sense and we have corrected this.*

Line 65. It is unclear whether "consituents" here is referring to lipids (line 63) or "biochemical constituents" of line 61.

*Response: Agreed, and we have clarified this (LINES 67-70).*

Line 72. Is "temps" an accepted abbreviation of temperature?

*Response: We have corrected this (LINES 76-77).*

Line 75. "planktotrophic larvae, including those of decapod crustaceans".

*Response: We have simplified this text (LINES 80-81).*

Line 101. "often protracted egg development in H. americanus". When is embryo development not protracted? Do the authors mean to say that 11 months is protracted and 9 months is not (see line 93)?

*Response: This could be confusing and we have shortened this section and reworked the rest of the text (LINES 106-120).*

Line 102. How is this behavior "dramatic"? Replace qualifiers based on drama (lines 102, 251, 276, 312, 375 and 418) by something less theatrical: substantial, important, etc.

*Response: These qualifiers have either been deleted (LINE 102, 418) or replaced (e.g., LINE 277, 366).*

Lines 120-122. How are these studies contradictory? And rephrase "it is evident that egg resources influence their growth…": what is "their" referring to? I assume "egg" not "egg resources" is the subject, but the egg does not grow, it is the embryo within that does.

*Response: We have eliminated this section to shorten the overall manuscript.*

Lines 148-149. "in a large… tanks". One or several tanks?

*Response: This was a single tank and has been clarified (LINE 166).*

Lines 149-150. "Tanks were exposed to ambient light and sand-filtered seawater". The tanks are not "exposed" to water.

*Response: This has been clarified (LINES 166-170).*

Line 154. "These samples also served as covariates for all subsequent analyses". Samples cannot be covariates; however, variables measured on those samples may serve as covariates.

*Response: This is absolutely correct and was an error on our part. This statement has been eliminated from the text (LINES 177-178) as well as from our statistical analyses and did not affect our results.*

Line 155. "Only lobsters whose eye index was less than 18%". I believe you are referring to the eye index of lobster embryos.

*Response: Yes, that is correct. This has been fixed (LINE 174).*

Lines 169-170. Rephrase.

*Response: This section has been rephrased (LINES 191-195).*

Line 171. "Constant temperatures were chosen to simulate a favorable growth temperature similar to eggs observed in Mackenzie (1988)". Rephrase.

*Response: We have rephrased this a bit (LINES 135-136).*

Line 176. Rephrase as "to simulate seasonal temperature changes obtained from historical and real-time data published…"

*Response: This section has been rephrased (LINES 199-205).*

Line 217. "longest length": change to "longest axis". And since only the longest axis was used to calculate egg volume, this volume is certainly overestimated unless the eggs were perfectly spherical.

*Response: This is a good point. We have corrected this (LINE 407).*

Line 217. Can a "calculation" be "measured to the nearest 0.01 mm"?

*Response: This should be to the nearest 0.1 and has been fixed (LINE 407).*

Line 187. Rephrase "Rather than mechanically separate eggs, this technique was chosen for its efficacy".

*Response: We eliminated this phrase as it's not necessary (LINES 214-215).*

Line 277. "hatched sooner". No results or reference to support this statement is provided. Please specify: when in the 3 treatments did hatching occur?

*Response: This data was assessed in a companion study (Goldstein and Watson 2015) but we have removed this phrase to avoid confusion (LINE 315).*

Line 285-288. How does the Sibert et al. study suggest "a large influence in the rate of temperature change between inshore and offshore locations"?

*Response: Yes, that's a valid point. Sibert et al. (2004) state in their paper that during springtime there is a "dramatic increase in embryo biomass in a short time at the end of egg development led to the hypothesis that variations in water temperature under natural field conditions in late spring could have a great impact on the characteristics of hatching larvae, with potential consequences on their survival". Because this is only stated as a hypothesis in their paper, we went ahead to qualify that in our manuscript (LINE 255).*

Lines 302-303. Is this suggestion the authors' or that of the cited authors? Clarify.

*Response: This is the general paradigm found in studies that we have referenced.*

Lines 387-389. Rephrase.

*Response: We have done so in LINES 351-352.*

Line 405. "deterring"?

*Response: We have provided an alternative word choice ('modulate') (LINE 624)*

Lines 535-541. Jacobs et al. should come before Jaeckle.

*Response: Yes, that is right (LINE 762).*

**Reviewer-2 (Kiran Liversage)**

Basic reporting

The basic reporting is well done. The writing style was a pleasure to read and the authors have done a good job with referencing the related background literature.

*Response: Thank you for the positive feedback and we hope to improve things further.*

I will only make two points concerning the article structure:

1) I do not think the first two figures are really needed as they just show some photos of lab equipment.

*Response: This was also noted by the other Reviewer and we agree. Both Figure 1 and 2 have been removed.*

2) I do not see why figure 4 shouldn't be incorporated as a line graph into figure 3, as it has the same predictor variables.

*Response: As mentioned, we have re-configured our figures! Thank you for your advice and suggestion on this. We now have a total of three figures, one of which (temperature comparison graph suggested by Reivewer-1) has been added.*

Experimental design

I think there are some issues with the experimental design:

1) The main issue is pseudoreplication. According to Hurlbert's review on pseudoreplication, the setup used in this experiment is classified as "isolative segregation" which is stated as one of the "various ways in which the principle of interspersion can be violated" (see Figure 1, Hurlbert, S.H., 1984. Pseudoreplication and the design of ecological field experiments. Ecological monographs, 54: 187-211). Ideally each lobster would be housed in is own separate tank and thus the eggs would be independent replicates that are randomly interspersed among other replicates. I think the authors will need to argue how their experimental setup can be valid despite the lack of interspersion of their treatments. Potentially the authors may be able to demonstrate that conditions (besides temperature) were similar among the treatments despite the fact they were isolated.

*Response: Kiran, this is an excellent point and something we struggled with quite a lot in designing a similar study that is already published (Biological Bulletin 2015, 228:1-12). In that particular study, we were following egg development through to hatch but also assessing hatching duration of female lobsters exposed to disparate temperature treatments. That study had a similar design, however because there could have been issues with chemical metabolites from hatching influencing lobsters (literature to suggest this) in adjacent sections of the tank, we opted to isolate each lobster as it approached hatching to eliminate this potential artifact. In this case, we were attempting to track the egg development of lobsters at three different temperature regimes and sample eggs at intervals before hatching occurred. Also, it is important to note that, as experimental replicates, lobster eggs are endowed with very thick-layered egg casings that make them isolated from each other egg (with the exception of hatching events when the chorion essentially ruptures). So, we are confident that keeping females isolated within the same tank (given the question we asked and data we were collecting) was acceptable given the research question and associated analyses we sought to explore.*

*We are familiar with Hurlbert's study and always try and keep that mantra in-mind for experimental design, especially in the lab! However, besides the biological rationale, there were some very concrete, logistical reasons that we were not able to exercise a 'pure' replicated study that was flawless. The marine lab where these studies were conducted had a finite amount of space and husbandry resources, and we were not able to isolate each lobster on 15 separate incoming seawater lines. This would have been especially problematic for the offshore and constant seawater treatments where we were simulating these conditions with heater and chiller units. Instead, we created a reservoir tank where we could very accurately control these conditions for each treatment. Therefore, we tried very hard to minimize a lack of independence and feel that what we came up with, especially given the question we were trying to answer was a legitimate design for this study.*

2) I had trouble following some aspects of the methods descriptions. Line 155-156 states there were 2 tanks used per treatment, and there were 3 treatments, so I was expecting 6 tanks, but line 165 states there were 4 tanks. Also, having 4 tanks does not seem to fit with having a total of 15 lobsters. Thus I

was not able to work out how the experimental setup was organised, although it might become clear if the methods writing is changed/expanded and/or a diagram provided.

*Response: Yes, I see how this could be confusing. First, there was an oversight on our part as there were six tanks, not four (2 tanks per each of the three treatments: inshore, offshore, and constant) (LINES 128-132). Second, there are visual details of this setup and complete details in the Goldstein & Watson (2015b) publication: Biological Bulletin. 228: 1-12 and we made sure to reference this in the methods.*

3) For this experimental design to work, all the treatments should have had eggs with the same level of development at the start of the experiment. From figure 3 it does seem the lipids and proteins were very similar among treatments, but in Figure 5 it seems there may have been differences in size at the 1st month. I would ask that the authors check (e.g. ANOVA) that all egg variables were statistically similar among treatments when the experiment started.

*Response: Worth noting for sure and yes, we did check out initial variability here and they were statistically similar.*

4) ANOVA assumes homogeneity of data variance, which does not appear to have been tested (e.g. Cochran's test or Levene's test). It could also be argued that as only 15 lobster were used, the sample size is quite low in which case normality of distributions will also be an important assumption to test. I would ask that the authors please address these aspects of their statistics.

*Response: Kiran, we actually did run assumption tests in SAS as a pre-requisite to our ANOVA output. I clarified this point in the 'data analysis' section of the methods (LINES 193-194). It is true that the sample size was a bit on the low side, although I egg numbers and replicated that we tested were fine. Thank you for pointing this out.*

5) On lines 153-155 there is a brief comment that "A subset of the eggs in each clutch were viewed under a dissecting scope and staged... These samples also served as covariates for all subsequent statistical analyses". The use of covariates is never mentioned again in the Data Analysis section nor any results concerning covariates in the results section. If covariates are used then the analysis will be an ANCOVA (not ANOVA as stated on line 223). If the analyses really did have a covariate then I would ask that there is more description of why and how this was done.

*Response: This is quite right and we never really used the data in this manner. We have removed this confusing statement from the text since we did not use ANCOVA models in our analyses.*

6) In the lipid and protein analyses, the constant temperature treatment was missing the last data point because the eggs hatched, and thus the design is unbalanced (i.e. May data for only 2 out of 3 treatments); was the statistical procedure used able to handle an unbalanced design? It would be my preference to have finalized the whole experiment as soon as it becomes impossible to continue getting data from one of the treatments. Overall, the fact that the design was unbalanced needs to be discussed.

*Response: This is for sure a biological constraint as you point out, the eggs in the 'constant' temperature treatment are hatching sooner than the other two treatments, leaving no data for this treatment at the last sampling point in May. That said, we feel that this is one of those cases where the biological output (i.e., eggs hatching sooner) is informative by itself. However, from an experimental design point-of-view,*

*I suppose one could test, in part, the efficacy of an unbalanced design or perhaps the probability of a Type II error by calculating a power statistic (1-β) or the probability that we rejected a false null hypothesis. We do not feel that this is warranted here, but if this a major sticking point, we can calculate this and report on it in our Results section.*

7) I would say that the regression analyses (Figure 4) are not needed, as the significant effect of time and the pairwise tests in the ANOVA analyses provide all this information.

*Response: We agree and have removed this figure in favor of leaving Figure 2 and the associated pairwise test table.*

Validity of the findings

I would say that the validity of the findings is not certain at this stage due to potential issues with pseudoreplication and analysis assumptions, and the authors will need to do more convincing for the reader that their conclusions are valid.

*Response: We've discussed this earlier – please see above comments.*

Comments for the Author

Overall, if efforts were made to present this study in a concise way then it could have only 1 figure of results and perhaps 1 table. The discussion is well written and full of background information, but much of this is loosely related to the topic at hand and if this were reduced then the manuscript could probably only be considered as a short communication. Although it may be possible to work through most of the issues about experimental design, given that there are quite a few of them, and that the volume of results is quite low, I would be hesitant to support this manuscript without substantial and possibly fundamental changes.

*Response: I think there are some good points made here and we have made the effort to scale back some of the figures and text. We have eliminated Figures 1, 2, and 4 from the original manuscript as these are good recommendations and help to eliminate redundancy and excess. By clarifying things even more we hope that, collectively, these improvements will make this manuscript more clear, concise, and effective.*

Here are a few further minor comments:

1) It would be very useful to see statistics tables and these will help the reader work out how the experiment was set up and the analyses done.

*Response: We agree and have attempted to integrate some of these summary stats into our Results section as well as in the two Tables that are included. We feel that these approach provides an adequate representation of our design and analyses performed.*

2) Line 221: Here you mention sampling of egg volumes was done at 5 times but Fig. 5 shows only 4 sampling times. Also on Fig. 5, why are the months different to the sampling months on the other graphs?

*Response: This is good thing to point out and we did (unfortunately) sample eggs for volume at offset temporal intervals. We did this because we did not initially intend to examine egg volume but decided to do so after egg sampling for biochemical assays had already started. Additionally, we had no intentions*

*to try and match up egg volume with biochemical data. Still, the four sampling intervals that we did choose (Oct, Nov, Jan, Mar, May) do encompass a temporal sampling window reflecting a full season of growth, which one could argue represent a full season of growth. Therefore, we have attempted to clarify this in the manuscript (LINES 174-179).*

3) Please put SE bars on all data graphs.

*Response: Duly noted.*

4) I think all ecological laboratory studies should have at least some discussion/mention about the limitations of this approach. With enough effort it would be possible to measure lobster egg development in the field which would provide much higher quality results.

*Response: This is probably worth mentioning for sure, and we actually did test this method in the field in Goldstein and Watson (2015b). In this scenario, we held egg-bearing lobsters in cages both inshore and offshore locations in the field, and we sampled their eggs at regular intervals, comparing those eggs with those being held concurrently in the laboratory – we saw no significant differences in development rate between lab- or field-incubated eggs. For following biochemical egg attributes as we did in this study, I would not expect to see stark differences between field and lab conditions, but your point is well-taken and we have made a brief statement to this effect in the 'Discussion' section (LINES 237-243).*

---

## Round 0.3 · accepted · Accept

You have dealt with the referees comments adequately. I look forward to seeing further research in this area and perhaps evidence that, as you have suggested may occur, lobster behaviour is tailored to deal with match-mismatch in ecological conditions.

#

---

## Author Rebuttal · Round 0.3

March 20, 2019

PeerJ, Inc.
PO Box 910224
San Diego, CA 92191
email: authors@peerj.com

Dear PeerJ Editorial Staff,

We are pleased to resubmit to you and your editorial staff our newly revised manuscript (Article ID 25864), 'Biochemical changes throughout early- and middle-stages of embryogenesis in lobsters (*Homarus americanus*) under varying thermal regimes', for re-assessment and hope that you and the two Reviewers find our improvements acceptable for approval.

Since this time we have been able to fully address both Reviewers' follow-up comments and suggestions, and have corrected, clarified, or improved just about all these issues that are detailed in a point-by-point response. We have further consolidated Figures that were not too informative and improved one as well. We have fixed and/or clarified ambiguous writing, and qualified some of our statements as well. We are providing a marked-up version of the manuscript and a 'clean' version so that all changes that were made can be seen.

Finally, we have addressed all the mandatory corrections and formatting issues brought to our attention by the editorial staff at PeerJ – Thank You.

We hope that you will find our improved manuscript as a successful contribution to PeerJ. If you have any questions, please do not hesitate to contact me.

With Kind Regards,

Jason S. Goldstein, PhD.
Corresponding Author

**PeerJ Article ID: 25864**
**Reviewer Comments and Responses_2 (Minor Revisions)**

Article Title: Review of "Biochemical changes throughout early- and middle-stages of embryogenesis in lobsters (*Homarus americanus*) under varying thermal regimes" by J.S. Goldstein and W.H. Watson III, submitted to PeerJ.

**Reviewer-1**

The authors have now provided more details on methods and findings. I have only a few last comments.

Major comments:
1. Lines 123-125. "Only lobster embryos with eye indices less than 18%... were used for this study…". (1) This sentence has been modified in response to a comment in the first review of the manuscript. As it is now formulated, the sentence can be interpreted as meaning that the <18% eye index criterion was applied to embryos at all sampling times throughout the study rather than only on the first sampling occasion (5 Sept.). Please clarify by changing "were used for this study" to "were used at this sampling time". (2) What were these early-sampled embryos used for? They do not appear to have been used for biochemical determinations nor for volume measurements (although another sampling for volume determinations by treatment was done in mid-September) and no information on them is reported in the results section.

*Response: We agree and have rephrased this to reflect this (LINES 123-126). In regards to the second inquiry, we did this initial assessment of lobster eggs to ensure that we did not use lobsters whose embryos were too far in their development to diminish our results or initial data for obtaining baseline measurements.*

2. Lines 202-207. Maybe it's worth noting that temperatures in the inshore and offshore treatments converged quickly at the outset of the experiment and diverged markedly starting only in early April (Fig. 1). Thus it is not so surprising that there was no difference in embryo development between inshore and offshore treatments, at least until the March sampling. It is perhaps more surprising that the late May sampling did not show a greater difference in biochemical content between the warmer inshore (≈9-10°C) and colder offshore (≈4°C ) treatments after almost 2 months of divergent temperature.

*Response: This is a good point, especially since temperatures over the course of the study were not significantly different. They did, however, diverge very much so in the spring which was a likely cause of differential egg development. Reviewer-2 also had a similar comment, and so we have inserted some modified text in LINES 211-214, and we also touch on this in the discussion with respect to biochemical differences.*

3. Lines 330-331. "up until Stage IV (post-larval), lobsters depended upon stored capacities of lipids". But are the larvae not also feeding exogenously, as stated in the introduction?

*Response: We have clarified this to reflect the fact that the Stage IV phase in lobsters is a rather transitionary one, during which they obtain nutrition from both stored reserves and active raptorial feeding.*

4. Line 350. Subsection on "Female size and condition". Neither of these variables was considered in this study, and so this subsection could be removed or substantially reduced in length.

*Response: This is a valid point so we have removed text in this sub-section to give it a more focused approach.*

Minor comments:
Line 56. "for their development".

*Response: We have revised this whole sentence for clarity.*

L59. "Lipids comprise provide the structural".

*Response: This has been corrected.*

L62. "proteins as are the basic".

*Response: This has been corrected.*

L71-72. Lobster larvae do derive their nutrition from exogenous and endogenous sources, but this is not true of all crustacean larvae (some are only lecithotrophic). Can be fixed by: "because in most species they derive".

*Response: This is a good point and we have reworked this sentence.*

L120. "to the propus".

*Response: This has been corrected.*

L158. "a fine powder".

*Response: Thank you for seeing this; this has been corrected.*

L183-184. "All calculations eggs were measured to the nearest 0.1 mm".

*Response: This has been corrected.*

L334. In line with a comment in the review of the first version of the manuscript, please consider changing "The increase of water in the eggs (egg volume) as seen in this study and others is directly related to water uptake and has been noted to increase by more than 50% over the course of development (Pandian, 1970) " to "The increase in egg volume as seen in this study and others is directly related to water uptake and may reach more than 50% over the course of development (Pandian, 1970)".

*Response: This is a good suggestion and we have changed this accordingly (LINES 339-340).*

L359. "these factors in more depth. Since, since female size".

*Response: This has been corrected.*

**Reviewer-2 (Kiran Liversage)**

Basic Reporting
1. Line 58, 59: words "both" and "alike" maybe not needed

*Response: This has been corrected.*

2. Line 62: change "as" to "are"
*Response: Grammatically speaking, this is correct as is.*

3. Line 66: remove word "also"

*Response: This has been corrected.*

4. Line 88: I would say the word "lab" may be too colloquial

*Response: Agreed, and this has been modified.*

5. Line 102: change "we sought to test" to "we tested"

*Response: This has been corrected.*

6. Line 150: maybe to increase succinctness remove "and placed in labeled plastic sample trays".

*Response: Good point; this has been corrected.*

7. Line 211: maybe remove "very"

*Response: This has been corrected.*

8. Figure 2: please add standard error bars to the sampling points on the line graphs

*Response: This has been amended to Figure 2.*

9. Line 217: change "and these results are summarized in Tables 1 & 2" to "(Tables 1 & 2)"

*Response: This has been corrected.*

10. Table 1: the means in this table appear to be exactly duplicating the information on Fig 2. I think the standard errors and post-hoc tests should be presented on Fig. 2 and Table 1 removed.

*Response:* Yes, this could be considered redundant. We have eliminated Table 1 and integrated some of the caption material into the Figure 2 caption description for clarity (see Figure captions). Accordingly, we have also re-numbered the Tables in the text.

11. Table 2 legend: please mention in the 1st sentence that the values shown are P values.

*Response: This has been corrected.*

12. Line 236: change "The main goal of this study was" to "Our main goal was"

*Response: This has been corrected.*

13. Line 267: change "couple" to "two"

*Response: This has been corrected.*

14. Line 272: The use of the word "real" here is a bit confusing, I would revise.

*Response: Agreed. We removed this word and changed the sentence a bit (LINE 278).*

15. Line 305-309: this long sentence could be split in two.

*Response: A good point and we have changed this (LINES 310-314).*

16. Line 384: Change "dramatically" to "significantly"

*Response: This has been corrected.*

17. Figure 2, 3: It is confusing for the reader that bar graphs were used Fig. 3 and line graphs Fig. 2, when they both show very similar time series. I would recommend for consistency to change all time series as line graphs, unless there is some specific reason its like this.

*Response: Although we can appreciate the consideration being made here, we feel that the bar graph for egg volume is effective as is and we are inclined to keep this way.*

Experimental design
18. I think it would be useful for some of the discussion about pseudo-replication in the response to reviewers document to be briefly mentioned somewhere in the manuscript.

*Response: We have provided a discussion of this (LINES 153-163), as suggested.*

Validity of the findings
19. Line 102-104: The hypotheses should directly match the tests done (i.e. the actual statistical comparisons). The only factors tested here were temperature and time; no measurements of migration were made. Also, the temperatures were not changed according to any migrations, they just tested how eggs would develop from lobsters staying either inshore or offshore. So the hypotheses should state something like "It was hypothesized that different water temperatures, simulated in the laboratory to be similar to temperatures experienced by lobsters in varying coastal zones (inshore vs offshore), will affect egg biochemistry. This hypothesis was tested to gain information useful for understanding how egg development may be affected by lobsters migrating between those zones"

*Response: This is a fair and valid point and we have re-worked our hypothesis based on your suggestion, which is much more clear and straight-forward (LINES 103-107).*

20. Line 207: the lack of significant difference in temperature between inshore and offshore water was surprising and may influence the validity of the conclusions, as this study is meant to inform about lobsters migrating between these areas to expose their eggs to different temperatures. Please provide some text about the implications of the non-significant

temperature result in the discussion.

*Response: Yes, the **average** inshore vs. offshore temperature over the course of the study was not different. However, **the rate of change in temperature** between these two locations (Fig. 1) is substantial and enough to then modulate disparate changes in development and subsequent hatch as noted in Goldstein and Watson (2015a,b). In the revised Discussion we spend more time on this point.*

21. Line 324-326: I think this text needs to be clarified/corrected. I guess the "constant" temperature being referred to here is the one used in this experiment, which according to line 248 appears more optimal for growth, but here it is stated as "sub-optimal". Also, if more protein is used for energy as stated here, then there should be less rather than more protein in the "constant" treatment which is not the case (Fig. 2).

*Response: We dealt with this in the Discussion a bit and removed the phrase 'sub-optimal' which is misleading.*

22. To allow the reader to understand the conclusions better, please somewhere early in the discussion provide a clear statement about whether or not the results supported the specific hypotheses from the introduction.

*Response: We have addressed this in the Discussion section in various places.*

23. Line 384-385: Please mention here why "there is still the potential", rather than just saying that the potential exists. It seems the results may indicate that no evidence could be found to support the hypothesis that "variations in the energetics of embryogenesis influenced by the seasonal movements of some lobsters to and from these two disparate locations", especially since these disparate locations did not even have significantly different temperatures according to my understanding (line 207). Also I think the word "are" is needed between "embryogenesis" and "influenced".

*Response: We have also reconciled this in the Discussion.*

Comments for the Author
24. I think a concluding sentence in the abstract stating the relevance of the results would be useful.

*Response: We have revised the Conclusions section accordingly.*

25. Line 39: add "potentially" between "thereby" and "exposing", especially since this study did not appear to find any significant different (Line 207) between the two areas where migrations occur between.

*Response: This has been corrected.*

26. Line 130: Here it says the "constant" treatment was 12 degrees, but in figure 1 it seems this treatment had 16 degrees. Please correct one or the other.

*Response: Thank you for catching that mistake in the text. Should be 16 C and we have changed this.*

27. Line 211-217: According to how I understood this, here all the results are from the 2 factor analyses. Firstly the text here details the main effects, then results of the interactions. The interactions are all significant, which makes it redundant to mention the main effects as was done first (i.e. effects of the factors in isolation are not informative). If this understanding is correct, then the text from line 211-215 may not be needed.

*Response: This was a bit redundant and diluted the main thrust of the summary statistics, so we revised this section and remove much of the text in the original lines 211-215 (now LINE-216).*

28. Line 300-302: a reference needs to be provided for this statement.

*Response: This has been added in the text (LINE-307) and in the References section.*

29. Line 350-378: These paragraphs are on a different topic to the study and are sufficiently unrelated that they confuse the story a bit and I would say they are probably not needed, or could be reduced to a few sentences.

*Response: This same issue was also brought up by Reviewer-1 and we have addressed this by substantially revising and shorting this section.*

Editorial office
**Funding Statement:**
At the next revision, please use full names, instead of initials, for the author names in the Funding Statement. Edit here https://peerj.com/manuscripts/25864/declarations/#question_18.

*Response: We will do this.*

**Tables:**
Tables should not be an image pasted into the Word document. At the next revision, please revise Table 2 so that the text is editable and upload it as a separate Word document (one file per table), including a title and any necessary legends in the text fields using the Edit button to the right of the file name here https://peerj.com/manuscripts/25864/files.

*Response: Yes, we will take care of this.*

**Figures:**
A) Figure 2 has multiple parts. Each part needs to be labeled alphabetically to use (A, B, C, D, etc) instead of directions (left, right, upper, lower, etc). At the next revision, please provide a replacement figure measuring minimum 900 pixels and maximum 3000 pixels on all sides, saved as PNG, EPS, or PDF (vector images) file format without excess white space around the images.

*Response: This has been revised as indicated above by the editorial office.*